environmental chemistry/materials science/organic chemistry

polymers of intrinsic microporosity, spirobifluorene-based polymers of intrinsic microporosity, methylene blue, surfactants, water treatment

**Authors for correspondence:**
Entesar Al-Hetlani
e-mail: entesar.alhetlani@ku.edu.kw
Mariolino Carta
e-mail: mariolino.carta@swansea.ac.uk

This article has been edited by the Royal Society of Chemistry, including the commissioning, peer review process and editorial aspects up to the point of acceptance.

# Spirobifluorene-based polymers of intrinsic microporosity for the adsorption of methylene blue from wastewater: effect of surfactants

Entesar Al-Hetlani[1], Mohamed O. Amin[1],

C. Grazia Bezzu[2] and Mariolino Carta[3]

[1]Department of Chemistry, Faculty of Science, Kuwait University, PO Box 5969, 13060 Safat, Kuwait
[2]School of Chemistry, Cardiff University, Cardiff CF10 3AT, UK
[3]Department of Chemistry, College of Science, Swansea University, Grove Building, Singleton Park, Swansea SA2 8PP, UK

Owing to their high surface area and superior adsorption properties, spirobifluorene polymers of intrinsic microporosity (PIMs), namely PIM-SBF-Me (methyl) and PIM-SBF-tBu (*tert*-butyl), were used for the first time, to our knowledge, for the removal of methylene blue (MB) dye from wastewater. Spirobifluorene PIMs are known to have large surface area (can be up to $1100 \, \text{m}^2 \, \text{g}^{-1}$) and have been previously used mainly for gas storage applications. Dispersion of the polymers in aqueous solution was challenging owing to their extreme hydrophobic nature leading to poor adsorption efficiency of MB. For this reason, cationic (cetyl-pyridinium chloride), anionic (sodium dodecyl sulfate; SDS) and non-ionic (Brij-35) surfactants were used and tested with the aim of enhancing the dispersion of the hydrophobic polymers in water and hence improving the adsorption efficiencies of the polymers. The effect of surfactant type and concentration were investigated. All surfactants offered a homogeneous dispersion of the polymers in the aqueous dye solution; however, the highest adsorption efficiency was obtained using an anionic surfactant (SDS) and this seems owing to the predominance of electrostatic interaction between its molecules and the positively charges dye molecules. Furthermore, the effect of polymer dosage and initial dye concentration on MB adsorption were also considered. The kinetic data for both polymers were well described by a pseudo-second-order

model, while the Langmuir model better simulated the adsorption process of MB dye on PIM-SBF-Me and the Freundlich model was more suitable for PIM-SBF-tBu. Moreover, the maximum adsorption capacities recorded were 84.0 and 101.0 mg g$^{-1}$ for PIM-SBF-Me and PIM-SBF-tBu, respectively. Reusability of both polymers was tested by performing three adsorption cycles and the results substantiate that both polymers can be effectively re-used with insignificant loss of their adsorption efficiency (%AE). These preliminary results suggested that incorporation of a surfactant to enhance the dispersion of hydrophobic polymers and adsorption of organic contaminants from wastewater is a simple and cost-effective approach that can be adapted for many other environmental applications.

# 1. Introduction

Water is the most important compound for living organisms to carry out their day-to-day vital activities. However, the water quality has severely declined with the increase in industrialization, commercialization and environmental changes [1]. Of particular interest is textile wastewater discharge into water resources, these pollutants are characterized by their intense colour, high pH and large chemical oxygen demand, which can block reoxygenation and sunlight from aquatic life [2]. Therefore, textile pollution is a serious growing issue that is threatening all living creatures, public health, agricultural, fishing and other activities causing toxic, carcinogenic and mutagenic effects. Continuing efforts from both academia and the industry have been dedicated to minimize these ongoing threats such as physical adsorption [3], oxidation [4], biological treatment [5], photocatalysis [6] and others. Among them, physical adsorption is proved to be one of the most popular approaches, because it does not produce harmful byproducts and does not require specialized apparatus [7]. Furthermore, the adsorbent can be easily removed and recycled using magnetic separation [8], filtration [3] or centrifugation [9]. To achieve maximum removal of contaminants from wastewater, optimization of adsorption is important to achieve maximum elimination rate, a series of parameters such as the adsorbent dosage, contaminant concentration, and in some cases temperature and pH control are crucial [10]. Thus, the preparation of efficient porous materials for this application is in great demand, and several adsorbents have been reported including nanoparticles [11], carbonaceous matter [12], composites [8] and bioadsorbents [13].

Polymers of intrinsic microporosity (PIMs) are rigid, water-insoluble and highly porous materials. They are produced from both commercially available and synthetic monomers and have large surface area ranging from 300 to 1100 m$^2$ g$^{-1}$ [14–16]. They have been extensively used as powder adsorbents, films or fibres in different areas for gas capture [17], storage [18] and separation [19,20]; they are also known for their photocatalytic activity [21], efficient synthesis of nanoparticles [22] and electroanalysis [19]. In a previous work, PIM-1, which is the first and probably the best studied polymer of the PIMs family, proved to be very efficient for the selective removal of dyes from water and ethanol-based solutions, especially when it was chemically modified and thermally treated [23]. In another study, it was further functionalized to form amine containing fibrous membranes and used for the filtration of organic dyes in water [24].

Herein, to the best of our knowledge for the first time, two types of spirobifluorene-based PIMs, namely PIM-SBF-Me (functionalized with methyl groups) and PIM-SBF-tBu (functionalized with *tert*-butyl) were used as adsorbents for the removal of MB from wastewater. Firstly, the polymers were simply dispersed in an aqueous solution containing MB and magnetically stirred. However, their initial dispersion was too poor, probably owing to their extreme hydrophobic nature, so it was improved by mixing them with cationic, anionic or non-ionic surfactants. In addition, for a complete investigation aimed to find the best conditions for the wastewater treatment, the surfactant nature, its concentration, polymer dosage and the MB concentration were optimized. Finally, the kinetic and isotherm data were investigated.

# 2. Experimental procedure

## 2.1. Chemicals and reagents

Methylene blue (MB), sodium dodecyl sulfate (SDS), Brij-35, cetyl-pyridinium chloride (CPC) were used as received. Commercially available reagents for the polymers synthesis were used without further

purification. Anhydrous dichloromethane and tetrahydrofuran were obtained from a solvent purification system. Anhydrous N,N dimethylformamide was bought from Sigma Aldrich. All reactions using air/moisture-sensitive reagents were performed in oven-dried or flame-dried apparatus, under a nitrogen atmosphere. The water used in all experiments was deionized water (DI) water from Elix Milli Q water deionizer.

## 2.2. Synthesis of PIM-SBF-Me and PIM-SBF-tBu

The synthesis of PIM-SBF-Me and PIM-SBF-tBu and their complete characterization are described in detail in [25]; their chemical structures are shown in the electronic supplementary material, scheme S1.

## 2.3. Instrumentation

To assess the adsorption and its kinetics ultraviolet-visible (UV-Vis) spectroscopy analysis was carried out using an Agilent Cary 5000 Scan UV–Vis–near-infrared spectrometer. The adsorption of MB on PIM-SBF-Me and PIM-SBF-tBu was investigated using Jasco FTIR-630 Fourier-transform infrared (FTIR) spectroscopy. The polymer, surfactant micelles and polymer-micelle size distribution was determined by photon correlation spectroscopy using a Zetasizer Nano ZS (Malvern Instruments, UK), with a frequency-doubled DPSS Nd: YAG laser with the output power of 50 mW operating at a wavelength of 532 nm and an angle of 175. All the measurements were performed at room temperature and for each sample, three measurements were carried out in disposable cuvettes. The refractive index and viscosity of the solvent were taken into consideration when measurements were taken.

## 2.4. Methylene blue adsorption experiments

The removal of MB from water using both PIM-SBF-Me and PIM-SBF-tBu was evaluated by performing a series of experiments. Initially, 1–10 mg of both polymers was added to 10 ml of 5.0–25.0 mg l$^{-1}$ MB solution in the presence of a surfactant. After mixing for 100 min, the remaining concentration of MB in the solution was determined by measuring the change $\lambda_{max} = 662$ nm in a quartz cuvette using a UV–Vis spectrometer. The adsorption efficiency (AE) for MB on both polymers was estimated using the equation below [8]:

$$\%\mathrm{AE} = \frac{(C_o - C_t)}{C_o} \times 100, \tag{2.1}$$

where $C_o$ and $C_t$ represent the initial MB concentration and the MB concentration at time $t$ (mg l$^{-1}$), respectively. The amount of MB adsorbed per unit mass of PIM-SBF-Me and PIM-SBF-tBu was calculated using the equation below:

$$q_t = \frac{(C_0 - C_t)V}{m}, \tag{2.2}$$

where $V$ and $m$ are the volume of solution (l) and the mass of the polymers added to the MB solution (g), respectively. At equilibrium, the equation can be modified:

$$q_{eq} = \frac{(C_0 - C_{eq})V}{m}, \tag{2.3}$$

where $q_{eq}$ and $C_{eq}$ are the amount of MB adsorbed per gram of the polymer (mg g$^{-1}$) and the concentration of MB at equilibrium (mg l$^{-1}$), respectively.

All adsorption isotherms and kinetic measurements were performed using the optimized conditions of adsorbent dosage and initial dye concentration. In brief, the optimum amount of the polymer was added into 10 ml of MB dye solution. The mixture was magnetically stirred, and the concentration of dye in the solution was determined at definite time intervals and analysed spectrophotometrically without filtration.

## 2.5. Reusability of the polymer

The reusability of the polymer was conducted by separating the adsorbed MB dye from solution. After performing the adsorption experiment, the MB–SDS–polymer mix was washed with acetone, as its

**Table 1.** Surfactants, their concentrations and %AE used in the removal of MB from water using PIM-SBF-tBu.

| surfactant | concentration (mM) | %AE |
|---|---|---|
| CPC [27] | 0.06 (<CMC) | 6 |
| | 0.12 (CMC) | 13 |
| | 0.36 (>CMC) | 5.2 |
| Brij-35 [28] | 0.045 (<CMC) | 0.9 |
| | 0.09 (CMC) | 46.8 |
| | 0.27 (>CMC) | 31.8 |
| SDS [29] | 4.2 (<CMC) | −s |
| | 8.4 (CMC) | 99.7 |
| | 25.2 (>CMC) | 99.7 |

addition caused the separation of SDS and MB from the polymer, owing to the poor solubility of SDS in acetone. Thereafter, the mixture was subjected twice to ultrasonication with DI water for 30 min followed by filtration. Finally, the polymer was collected and dried in oven at 80°C overnight. The reusability study was performed by mixing the recovered dried polymer with a fresh solution of the MB dye and the SDS surfactant, as described in §2.4. Using this protocol, the adsorbent was regenerated and recycled three times in the adsorption experiments.

# 3. Results and discussion

## 3.1. Effect of surfactant nature and concentration

MB is a well-known cationic dye with maximum absorbance in the visible region at 662 nm and a shoulder at 612 nm [26]. Initially, PIM-SBF-Me and PIM-SBF-tBu were mixed with MB solution and magnetically stirred; however, the dispersion of the polymers was poor and kept separating from the aqueous solution. After mixing, the recorded adsorption efficiencies (AEs) were 68.1% and 9.6% for PIM-SBF-Me and PIM-SBF-tBu, respectively. This poor performance was related to the hydrophobic nature of the polymers, which limited their dispersion in the aqueous solution and, subsequently, the adsorption of the dye.

Therefore, surfactants were employed to improve the dispersion of the polymers in aqueous solution. Surfactants are molecules that can form self-assembled clusters known as micelles, and are commonly used to form emulsions between water and oil. In this work, surfactants were used to enhance the dispersion of the extremely hydrophobic polymers in the aqueous solution of MB. Three types of surfactants were tested with the proposed polymers at concentrations less, equal and higher than their corresponding critical micelle concentration (CMC) (table 1). The chosen surfactants, CPC (cationic), SDS (anionic) and Brij-35 (non-ionic), were added to the polymer(s) and the MB mixture at concentrations one-half CMC, CMC and 3× CMC. Initially, UV–Vis absorbance was recorded for the surfactants and MB solution mixture (without polymer). It was noted that, when MB was mixed with SDS at concentration lower than SDS CMC, the solution changed in colour and the absorbance performance also changed [30]. This behaviour is well known for MB which is attributed to the formation of a highly aggregated dye–surfactant pair generated by the electrostatic attraction between SDS and MB [31]. For this reason, the sample was discarded and no data were collected for SDS below its CMC value.

Interestingly, the polymers were completely dispersed in MB aqueous solution using the three surfactants at all tested concentrations. These solutions were magnetically stirred and their absorbance was recorded and their %AE are shown in table 1 and an example of the UV–Vis absorbance spectrum obtained for MB using PIM-SBF-tBu as adsorbent in the presence of different types of surfactants is shown in figure 1.

Poor MB adsorption was observed using cationic surfactant; good results were obtained with non-ionic surfactant at and above CMC, whereas superior adsorption efficiency of 99.7% was obtained with anionic surfactant. Thus, the anionic was the chosen surfactant for the experiments.

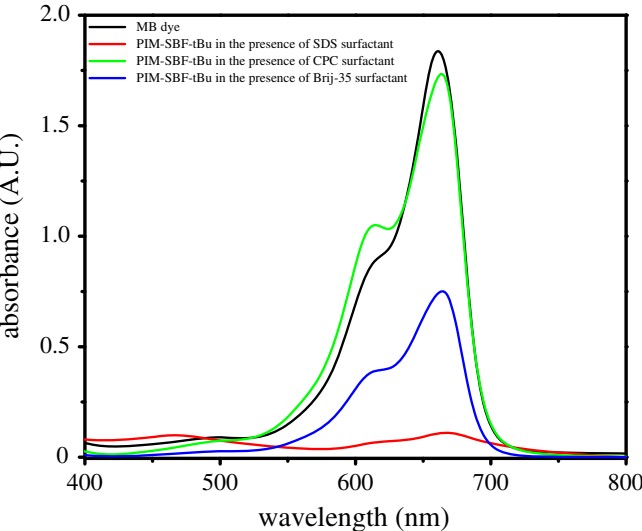

**Figure 1.** UV–Vis spectra of MB before and after mixing with PIM-SBF-tBu in the presence of SDS, CPC and Brij-35 at CMC.

## 3.2. Effect of PIM-SBF-Me and PIM-SBF-tBu dosage

The dosage of both PIM-SBF-Me and PIM-SBF-tBu was investigated over a range of 1–10 mg using a constant concentration of MB (10 ppm) in the presence of SDS at its CMC. The absorbance of MB was recorded every 15 min without filtration and the AEs were assessed as shown in the electronic supplementary material, figure S1a,b. Interestingly, a high AE for PIM-SBF-Me was observed (approx. 97.5%) upon using only 1 and 3 mg of the polymer, and reached almost 100% when 5 mg was used. However, at higher adsorbent dosage 7 and 10 mg, the AE slightly dropped to approx. 94.5 and 93.9%, respectively. This behaviour can be ascribed to the formation of aggregations at high adsorbent dosage which subsequently decreased the number of sites available for MB adsorption [32]. Additionally, a similar trend was observed with PIM-SBF-tBu in which 78% AE was obtained using 1 mg of the polymer, which increased to 98.8% upon using 5 mg dose of the polymer followed by a slight decrease using higher adsorbent dosage. Thus, 5 mg adsorbent dose was decided to be the optimum polymer dosage, as illustrated in figure 2a,b, and therefore was used in further experiments.

## 3.3. Effect of initial dye concentration

The effect of the initial dye concentration on the AE of PIM-SBF-Me and PIM-SBF-tBu as a function of time was investigated, as shown in the electronic supplementary material, figure S2 and figure 3a,b. Notably, the AE of PIM-SBF-Me remained constant at 5 and 10 ppm of MB with a maximum of 100% after almost 60 min. However, upon using higher MB dye concentration, the maximum AE slightly decreased with a minimum of 99.3%. Similarly, the AE of PIM-SBF-tBu increased from 93.5 to 98.8% using 5 and 10 ppm MB dye, respectively, and slightly decreased at higher concentration of the MB dye.

Generally, for a given adsorbent, the number of the adsorption sites is constant; thus, at lower MB concentration, a good portion of active and accessible adsorption sites on both polymers are available during the adsorption process. However, at higher MB concentration, most of these sites are occupied, which causes a reduction in the available number [33]. It was, therefore, decided to use 10 ppm of the MB dye as the optimum concentration.

## 3.4. Adsorption kinetic

For both PIM-SBF-Me and PIM-SBF-tBu, the amount of MB adsorbed ($Q_t$) during a period of 100 min was investigated, as illustrated in figure 4. A rapid removal of MB dye by both polymers was observed after the first 15 min, owing to the quick accessibility of active adsorption sites, thereafter a state of equilibrium at which $Q_t$ plateaued was attained after 60 min, which was then used as the equilibrium time.

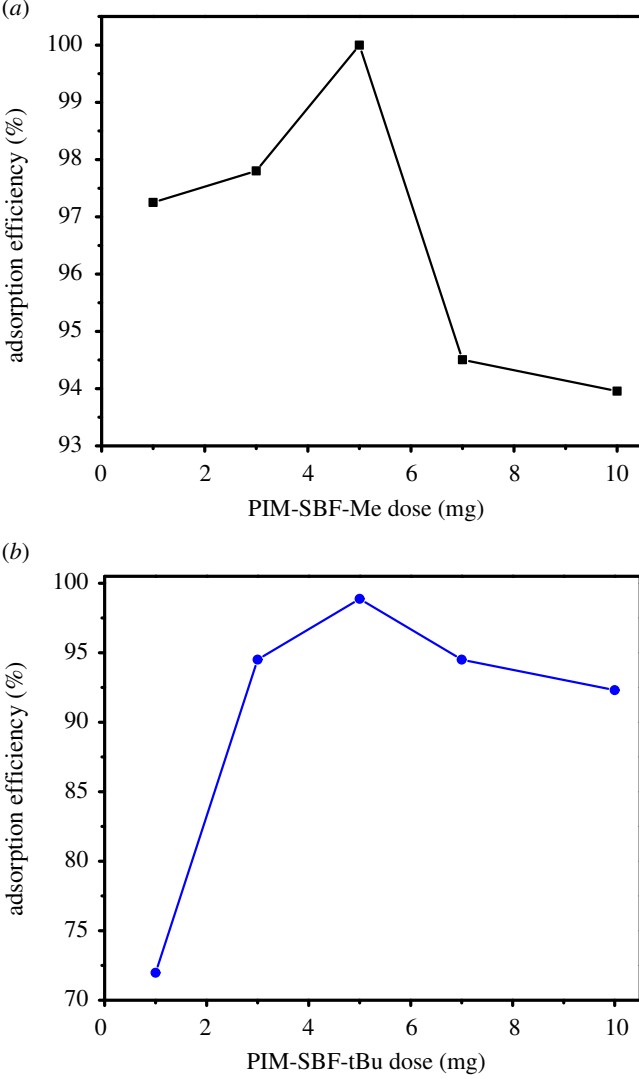

**Figure 2.** The adsorption efficiency versus (a) PIM-SBF-Me dosage and (b) PIM-SBF-tBu dosage. Conditions: MB initial concentration 10 ppm, temperature 25°C and SDS at CMC.

To further comprehend the features of adsorption process, both pseudo-first- and pseudo-second-order kinetic models were employed to fit the obtained experimental data. Both kinetic models are expressed in linear forms using equations (3.4) and (3.5), respectively:

$$\ln(Q_e - Q_t) = \ln Q_e - k_1 t, \tag{3.4}$$

$$\frac{t}{Q_t} = \frac{1}{Q_e}t + \frac{1}{k_2 Q_e^2}, \tag{3.5}$$

where $Q_e$ and $Q_t$ are the adsorbed quantity of MB dye at equilibrium and time $t$ (mg g$^{-1}$), respectively; and $k_1$ and $k_2$ are the rate constants for the pseudo-first-order (min$^{-1}$) and pseudo-second-order reactions (g mg$^{-1}$ min$^{-1}$), respectively.

The kinetic parameters and correlation coefficients ($R^2$) for both models were attained from straight-line plots, and the data are provided in table 2. Pseudo-second-order model showed great linearity with correlation coefficients of 1.00 and 0.999 for PIM-SBF-Me and PIM-SBF-tBu, respectively, indicating that the pseudo-second-order model is suitable to describe the obtained kinetic data for both polymers.

## 3.5. Adsorption isotherm

Studies of isothermal adsorption were employed to gain a better understanding of the interaction between MB and both polymers, and to determine the adsorption capacity at equilibrium ($q_e$). For this

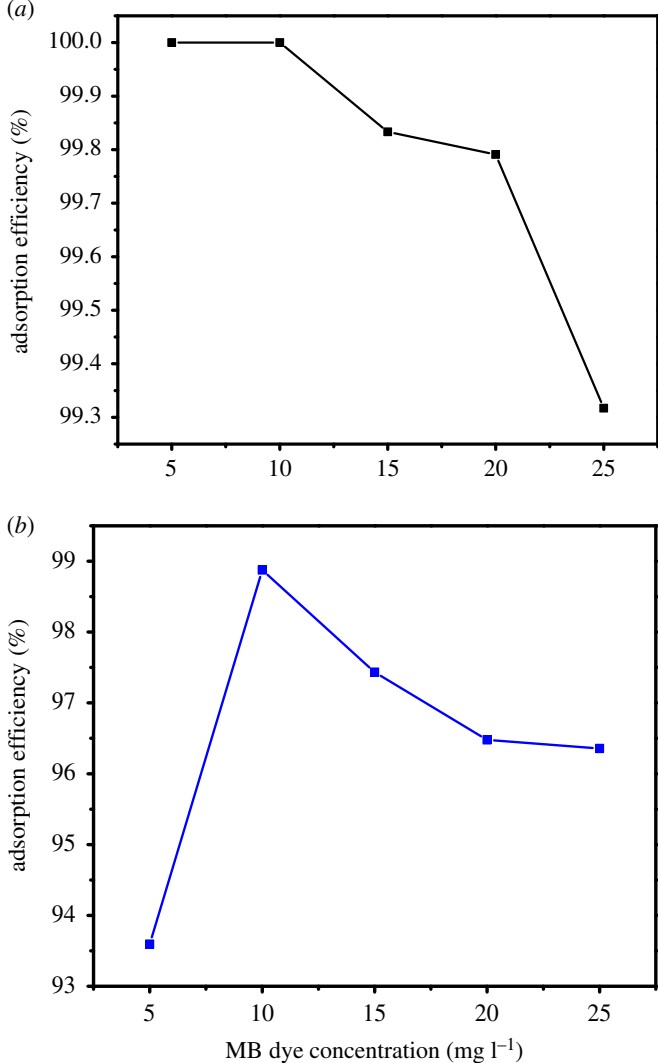

**Figure 3.** Effect of initial MB concentration on AE using (*a*) PIM-SBF-Me and (*b*) PIM-SBF-tBu. Conditions: polymer dosage 5 mg, temperature 25℃ and SDS at CMC.

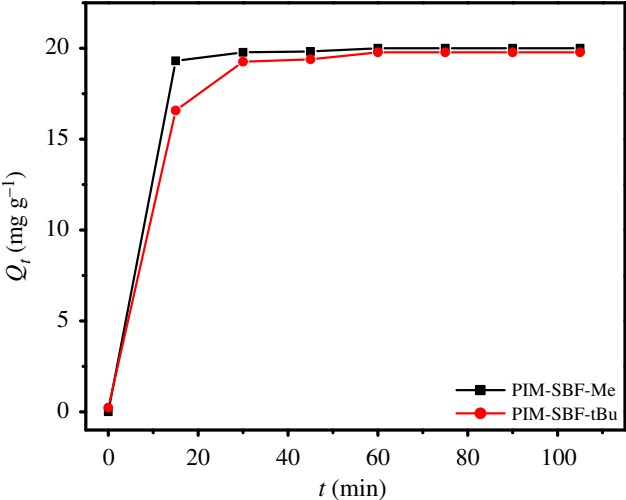

**Figure 4.** Effect of contact time on the adsorption rate of MB on PIM-SBF-Me and PIM-SBF-tBu. Conditions: MB initial concentration 10 ppm, polymer dosage 5 mg, temperature 25℃ and SDS at CMC.

**Table 2.** Kinetic parameters for the adsorption of MB dye on PIM-SBF-Me and PIM-SBF-tBu.

| adsorbent | pseudo-first-order model | | | pseudo-second-order model | | |
|---|---|---|---|---|---|---|
| | $k_1$ (s$^{-1}$) | $Q_e$ (mg g$^{-1}$) | $R^2$ | $k_2$ (g mg$^{-1}$ min$^{-1}$) | $Q_e$ (mg g$^{-1}$) | $R^2$ |
| PIM-SBF-Me | 0.17 | 24.24 | 0.831 | 0.15 | 20.08 | 1.00 |
| PIM-SBF-tBu | 0.42 | 655.0 | 0.799 | 0.037 | 20.04 | 0.999 |

**Table 3.** Fitting parameters of the isotherm models for the adsorption of MB on PIM-SBF-Me and PIM-SBF-tBu.

| adsorbent | Langmuir isotherm model | | | Freundlich isotherm model | | |
|---|---|---|---|---|---|---|
| | $K_L$ (ml g$^{-1}$) | $Q_m$ (mg g$^{-1}$) | $R^2$ | $K_F$ (ml g$^{-1}$) | $n$ | $R^2$ |
| PIM-SBF-Me | 11.9 | 84.0 | 0.988 | 71.76 | 5.11 | 0.976 |
| PIM-SBF-tBu | 1.06 | 101.0 | 0.953 | 46.14 | 2.53 | 0.982 |

purpose, Langmuir and Freundlich models were used to fit the obtained data from the batch experiments. The Langmuir isotherm is a theoretical model which assumes a monolayer coverage of adsorbate on a homogeneous adsorbent surface where interaction between them is negligible [34,35]. This model can be expressed as follows:

$$\frac{C_e}{q_e} = \frac{C_e}{Q_m} + \frac{1}{K_L \, Q_m},$$ (3.6)

where $C_e$, $q_e$, $Q_m$ and $K_L$ are the concentration of MB at equilibrium (mg l$^{-1}$), the amount adsorbed at equilibrium (mg g$^{-1}$), the maximum adsorption capacity (mg g$^{-1}$) and the adsorption equilibrium constant (l mg$^{-1}$), respectively.

The Freundlich isotherm model, instead, assumes multilayer adsorption over a heterogeneous adsorbent surface with strong interactions between the adsorbent and adsorbate. The Freundlich isotherm model can be expressed as follows:

$$\ln \, Q_e \, = \, \ln K_f \, + \, \frac{\ln C_e}{n},$$ (3.7)

where $K_f$ is the Freundlich constant, which is related to the sorption capacity, and $n$ is the adsorption intensity (or heterogeneity). Thus, to obtain the Langmuir parameters, a plot of $C_e/q_e$ versus $C_e$ was employed, whereas to obtain the Freundlich parameters, a plot of $\ln q_e$ versus $\ln C_e$ was used. Both models were employed to fit the obtained experimental data for MB adsorption onto PIM-SBF-Me and PIM-SBF-tBu and the results are shown in table 3.

The adsorption isotherm of MB dye onto PIM-SBF-Me and PIM-SBF-tBu is shown in figure 5. The findings showed that the linear correlation coefficients of Langmuir and Freundlich were 0.988 and 0.976 for PIM-SBF-Me, respectively, while for PIM-SBF-tBu, they were 0.953 and 0.982, respectively. That means, the Langmuir model better simulated the adsorption process of MB dye on PIM-SBF-Me, while the Freundlich model appears more suitable for PIM-SBF-tBu. Moreover, the maximum adsorption capacity calculated using the Langmuir model were 84.0 and 101.0 mg g$^{-1}$ for PIM-SBF-Me and PIM-SBF-tBu, respectively. The higher adsorption capacity of PIM-SBF-tBu compared to PIM-SBF-Me can be attributed to its larger surface area (882 m$^2$ g$^{-1}$) with respect to PIM-SBF-Me (752 m$^2$ g$^{-1}$).

Additionally, the Freundlich constant $(1/n)$ is used to describe the feasibility of the adsorption process. Commonly, if values of $n$ are between 2 and 10, it indicates good adsorption, 1–2 moderate to difficult adsorption, whereas less than 1 means adsorption is poor [36]. The values of $n$ for PIM-SBF-Me and PIM-SBF-tBu were 5.11 and 2.53, respectively, confirming the ease of MB to be adsorbed onto the prepared polymers.

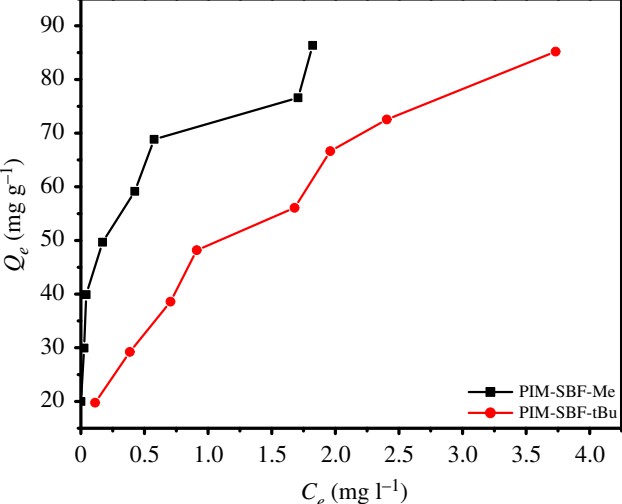

**Figure 5.** Adsorption isotherm of MB onto PIM-SBF-Me and PIM-SBF-tBu. Conditions: MB initial concentration 10 ppm, polymer dosage 5 mg, temperature 25℃, SDS at CMC and at 60 min contact time.

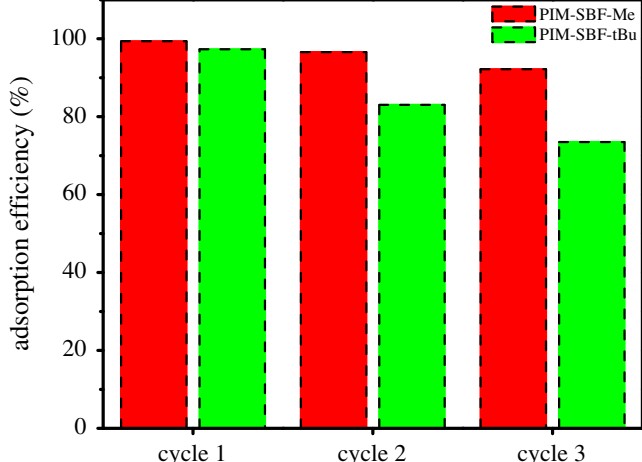

**Figure 6.** Reusability experiments of PIM-SBF-Me and PIM-SBF-tBu for removing MB from wastewater. Conditions: MB initial concentration 10 ppm, polymer dosage 5 mg, temperature 25℃, SDS at CMC and at 60 min contact time.

## 3.6. Reusability of PIM-SBF-Me and PIM-SBF-tBu

The reusability and stability of both polymers were studied by performing three adsorption cycles, as shown in figure 6. The experimental conditions were maintained the same in each cycle. Clearly, the removal efficiency of PIM-SBF-Me decreased from 99.38 to 96.57% and 92.18% in cycles 2 and 3, respectively, demonstrating no significant loss of AE (less than 8%). Conversely, a decrease in the AE of PIM-SBF-tBu was observed after three cycles (less than 25%), suggesting that PIM-SBF-Me has a higher recyclability over a period of use, compared to PIM-SBF-tBu.

## 3.7. Proposed adsorption mechanism and interactions

The mechanism that governs the adsorption of MB dye by the polymers is normally influenced by the interactions between the adsorbent and the adsorbate; however, the presence of surfactant as a dispersing agent showed a dramatic change in the adsorption process and consequently the interactions involved. Thus, to gain a better understanding of the adsorption process and to deduce the mechanism that governs the uptake of MB dye by the polymers, FTIR spectra were obtained for

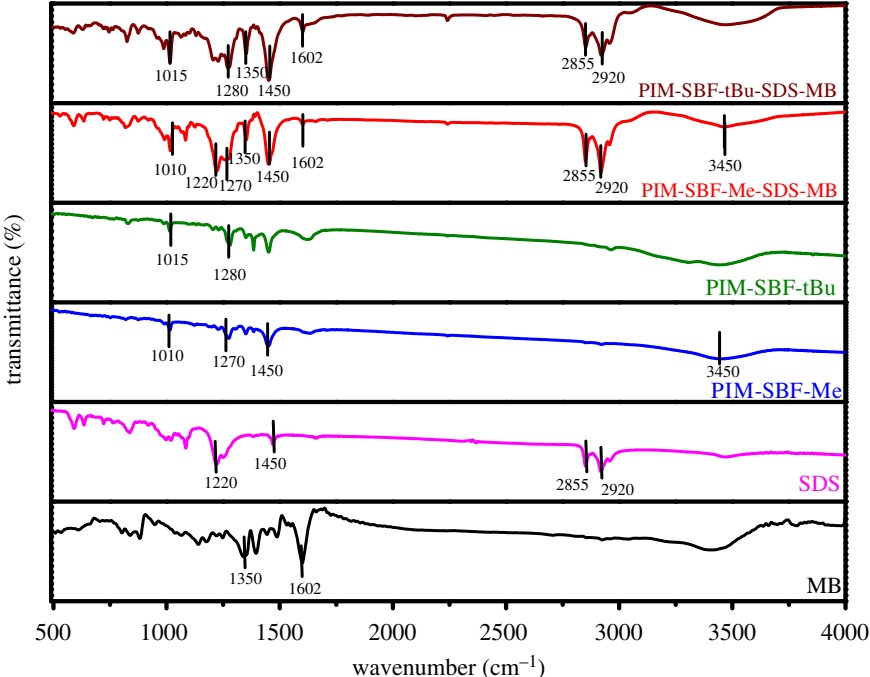

**Figure 7.** FTIR spectra of MB, SDS, PIM-SBF-Me, PIM-SBF-tBu, PIM-SBF-Me-SDS-MB and PIM-SBF-tBu-SDS-MB.

**Table 4.** Zetasizer measurements of SDS micelle with and without the presence of the polymers.

| material | size (nm) |
|---|---|
| SDS | 4.85 |
| SDS + PIM-SBF-Me | 142 |
| SDS + MB + PIM-SBF-Me | 250 |
| SDS + PIM-SBF-tBu | 142 |
| SDS + MB + PIM-SBF-tBu | 270 |

PIM-SBF-Me, PIM-SBF-tBu, MB and SDS before and after the adsorption, as illustrated in figure 7. Evidently, upon the adsorption process, the individual peaks of PIM-SBF-Me, PIM-SBF-tBu, MB and SDS were clearly observed with indistinguishable shift, which corroborates the physical nature of the adsorption process of MB by PIM-SBF-Me, PIM-SBF-tBu in the presence of SDS surfactant. Thus, the adsorption mechanism more likely took place in two steps; firstly, the dispersion of the hydrophobic polymers in the aqueous solution of MB aided by the surfactant molecules and secondly, the adsorption of MB dye on the polymer surface.

To further investigate the interaction between the polymers, dye and surfactant molecules, Zetasizer measurements were employed to measure the size of SDS micelle in the presence of the polymers and MB as shown in table 4. The micelle size in water is in similar to values previously reported in the literature [29]. However, upon the addition of the polymer, the size of the micelles increased, which could be owing to the solubilization of the polymer into the hydrophobic core of the micelles that may cause their expansion. The addition of MB to the mixture did not cause a significant increase to the micelle size in both polymers, which could be owing to the small size of MB molecules (or aggregates) that interact with the SDS charged head [37]. These changes in the micelle size proved the presence of interactions between the polymer and the surfactant, as illustrated in figure 8.

The adsorption of MB by the PIM-SBF-Me and PIM-SBF-tBu in the presence of anionic surfactants is most likely governed by hydrophobic interactions between the polymers and the surfactant hydrophobic tail. Additionally, different forces might be predominant between the dye and

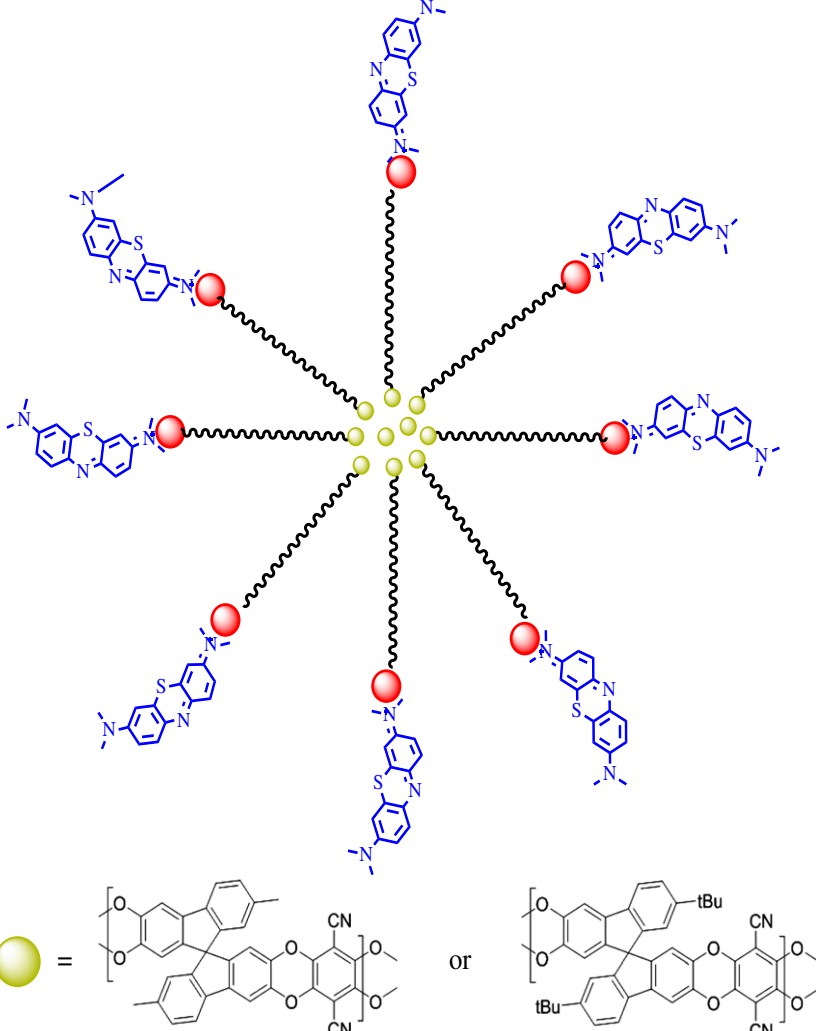

**Figure 8.** Proposed interaction scheme between PIM-SBF-Me (or PIM-SBF-tBu), MB dye and SDS surfactant.

surfactant, depending on the surfactant head group nature. Therefore, it is possible to note improved or diminished dye adsorption because of the surfactant head group. In the case of CPC, repulsion forces between the surfactant and dye molecules (both carry positive charges) were stronger than hydrophobic and van der Waals interactions, thus the interaction between the hydrophobic polymers and the dye was limited [38]. A better adsorption was observed in the case of the Brij-35, which could be related to the hydrophobic interaction between the polymer, MB and surfactant. However, the absence of stronger interaction between the polymer, MB and surfactant molecules limited the adsorption efficiency. The same type of interaction is observed with SDS; however, positively charged MB further interacted with SDS through the negatively charged sulfate moiety head group [37]. In this case, upon using an anionic surfactant, the adsorption of MB dye by the polymers was considerably enhanced owing to the electrostatic attraction between MB dye and the surfactant. MB monomers are solubilized inside SDS aggregations (micelles) through the charged head group [38], while the micelles hydrophobic tails interact with the hydrophobic polymers and thus improving the adsorption of MB on the polymers.

A comparison of experimental parameters recorded for the removal of MB dye from wastewater using a range of adsorbents, their dosage, equilibrium time and model, kinetic model and maximum adsorption capacity is shown in table 5. A small dosage was used for both polymers in a short period of time to achieve excellent adsorption capacities of 84 and 101 mg g$^{-1}$ for PIM-SBF-Me and PIM-SBF-tBu, respectively. This indicates that spirobifluorene-based polymers of intrinsic microporosity are good candidates for wastewater treatment application.

**Table 5.** Summary of experimental data obtained for the removal of MB dye from wastewater using a range of adsorbents, their dosage, equilibrium time and isotherm and kinetic models, and maximum adsorption capacity.

| sorbent | amount (mg) | equilibrium time | isotherm model | kinetic model | adsorption capacity (mg g$^{-1}$) | ref. |
|---|---|---|---|---|---|---|
| AC | 500 | 90 min | Langmuir | pseudo-second-order-kinetic | 47.62 | [39] |
| KOH-activated AC | 60 | 6 h | Langmuir | pseudo-second-order-kinetic | 250 | [40] |
| H$_2$SO$_4$-activated AC | 50 | 25 min | Langmuir | pseudo-second-order-kinetic | 62.5 | [41] |
| Fe$_3$O$_4$-AC | 10 | 6 min | Langmuir | pseudo-second-order-kinetic | 101.01 | [8] |
| Fe$_2$O$_3$-ZrO$_2$/Black cumin | 2000 | 2 h | Langmuir | pseudo-second-order-kinetic | 38.1 | [42] |
| Ag nanopartides loaded-AC | 15 | 4 min | Langmuir | pseudo-second-order-kinetic | 75.2 | [43] |
| potato (*Solanum tuberosum*) plant | 2000 | 25 min | Freundlich | pseudo-second-order-kinetic | 52.6 | [44] |
| surfactant-modified AC | 15 | 120 min | Langmuir | pseudo-second-order-kinetic | 232.5 | [45] |
| zeolite-AC | 100 | — | Freundlich | pseudo-second-order-kinetic | 143.47 at 30°C | [46] |
| rod-shaped manganese oxide (MnO) mixtures | 7 | 150 min | Langmuir | pseudo-second-order-kinetic | 154 | [47] |
| PIM-SBF-Me | 5 | 15 min | Langmuir | pseudo-second-order-kinetic | 84 | this work |
| PIM-SBF-tBu | 5 | 15 min | Freundlich | pseudo-second-order-kinetic | 101 | this work |

# 4. Conclusion and outlook

The dispersion of the hydrophobic polymers in aqueous solution for water-related applications can be cumbersome. For this reason, in this work, surfactants were employed to improve the dispersion of hydrophobic polymers PIM-SBF-Me and PIM-SBF-tBu in water, without the need for complex apparatus, high voltage or pH alteration. Anionic surfactant SDS showed enhanced dispersion of the hydrophobic polymers in water and assisted the adsorption process because of hydrophobic interactions with the polymer and electrostatic interactions with the positively charged MB dye. The results proved that the kinetic data for both polymers followed a pseudo-second-order model while the Langmuir model and Freundlich models seemed more suitable to describe the adsorption process of MB dye on PIM-SBF-Me and PIM-SBF-tBu, respectively. Furthermore, both polymers showed a relatively good maximum adsorption capacity of 84.0 and 101.0 mg g$^{-1}$ for PIM-SBF-Me and PIM-SBF-tBu, respectively. This simple approach can be further used for the removal of other types of contaminants from wastewater using hydrophobic adsorbent materials. Future work will involve studying the effect of pH and temperature on the adsorption process of different dyes in wastewater.

Data accessibility. All raw data, code, analysis files and materials associated with this study are available from the Dryad Digital Repository: https://dx.doi.org/10.5061/dryad.n2z34tmtq [48].

Authors' contributions. E.A.-H. and M.O.A. designed and performed the adsorption experiments, carried out the analyses and helped draft the manuscript; M.C. and C.G.B. prepared the polymers and completed their full characterizations and participated in drafting the manuscript. All authors gave final approval for publication.

Competing interests. We declare we have no competing interests.

Funding. This work was supported by the Kuwait Foundation for the Advancement of Sciences (KFAS-grant no. PN17-24SC-01).

Acknowledgements. The authors gratefully acknowledge the Kuwait Foundation for the Advancement of Sciences (KFAS-grant no. PN17-24SC-01) and Kuwait University Research Administration (KURA). RSPU facilities nos GS02/01 and GS01/05. Special thanks to KUNRF general facility no. GE 01/07 for the Zeta potential measurement.

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
