## [Reviewer comments · Royal Society Open Science]

Review History

RSOS-200741.R0 (Original submission)

Review form: Reviewer 1

Is the manuscript scientifically sound in its present form?

Yes

Are the interpretations and conclusions justified by the results?

Yes

Is the language acceptable?

Yes

Do you have any ethical concerns with this paper?

No

Have you any concerns about statistical analyses in this paper?

No

Recommendation?

Accept with minor revision (please list in comments)

Comments to the Author(s)

This is an interesting piece of work using state-of-the-art PIMs to remove dyes from water via physisorption. The characterisation techniques deployed here are complementary of each other. The language used here is suitable for scientific publication and the experiments here are very robust, leading to a solid conclusion of using PIMs as adsorbents to remove dyes from water.

I have the following questions in mind that remain unanswered in the manuscript:

- 1) As surfactants are used here as well, are they being adsorbed into the PIMs as well?
- 2) Did the authors determine the pore size distribution of the PIMs used here before and after MD adsorption and desorption? This information will be crucial to show the location of these dyes as well as the effect of surfactants on adsorption. I understand that adsorption of dyes is nearly impossible without surfactants here, but will the surfactants be adsorbed into the PIMs as well? I think it will be good if the authors can provide some gas adsorption analyses on such samples to provide a conclusive finding for these questions.
- 3) Did the authors measure the dye concentration in the solution after ultrasonication of PIM-dye mixtures? Will the amount of dye released in this solution be equivalent to the amount dye removed from the original MB/water mixture?

Overall, I think this paper is well-written and should be published after addressing these concerns.

Review form: Reviewer 2

Is the manuscript scientifically sound in its present form?

Yes

Are the interpretations and conclusions justified by the results?

Yes

Is the language acceptable?

Yes

Do you have any ethical concerns with this paper?

No

Have you any concerns about statistical analyses in this paper?

No

Recommendation?

Accept with minor revision (please list in comments)

Comments to the Author(s)

In this work, the author reported two spirobifluorene PIMs to remove the methylene blue (MB) dye from wastewater with the help of surfactants. The author explored the effects of different types and concentration of surfactants on the adsorption efficiency, and proposed the adsorption mechanism through reliable tests. The manuscript is well organized and has a great effect on the wastewater treatment. Therefore, I recommend this manuscript may be accepted for publication subjected to the incorporation for following minor corrections.

(1) Why does different chains on the benzene ring lead to the difference in adsorption performance, can the author simply explain this question from the structural perspective?

Decision letter (RSOS-200741.R0)

Dear Dr AlHetlani:

Title: Spirobifluorene-based polymers of intrinsic microporosity for the adsorption of methylene blue from wastewater: effect of surfactants
Manuscript ID: RSOS-200741

Thank you for submitting the above manuscript to Royal Society Open Science. On behalf of the Editors and the Royal Society of Chemistry, I am pleased to inform you that your manuscript will be accepted for publication in Royal Society Open Science subject to minor revision in accordance with the referee suggestions. Please find the reviewers' comments at the end of this email.

The reviewers and handling editors have recommended publication, but also suggest some minor revisions to your manuscript. Therefore, I invite you to respond to the comments and revise your manuscript.

Because the schedule for publication is very tight, it is a condition of publication that you submit the revised version of your manuscript before 19-Jul-2020. Please note that the revision deadline will expire at 00.00am on this date. If you do not think you will be able to meet this date please let me know immediately.

- 1) A text file of the manuscript (tex, txt, rtf, docx or doc), references, tables (including captions) and figure captions. Do not upload a PDF as your "Main Document".
- 2) A separate electronic file of each figure (EPS or print-quality PDF preferred (either format should be produced directly from original creation package), or original software format)
- 3) Included a 100 word media summary of your paper when requested at submission. Please ensure you have entered correct contact details (email, institution and telephone) in your user account
- 4) Included the raw data to support the claims made in your paper. You can either include your data as electronic supplementary material or upload to a repository and include the relevant doi within your manuscript

5) All supplementary materials accompanying an accepted article will be treated as in their final form. Note that the Royal Society will neither edit nor typeset supplementary material and it will be hosted as provided. Please ensure that the supplementary material includes the paper details where possible (authors, article title, journal name).

Kind regards,
Dr Laura Smith
Publishing Editor, Journals

On behalf of the Subject Editor Professor Anthony Stace and the Associate Editor Dr Chaohua Cui.

RSC Associate Editor:

Comments to the Author:

As a reviewer failed to submit the comments despite several reminders, another reviewer was consulted. I am sorry for the delay in getting back to you with the reviewers comments in this case.

RSC Subject Editor:

Comments to the Author:

(There are no comments.)

Reviewer comments to Author:

Reviewer: 1

Comments to the Author(s)

This is an interesting piece of work using state-of-the-art PIMs to remove dyes from water via physisorption. The characterisation techniques deployed here are complementary of each other. The language used here is suitable for scientific publication and the experiments here are very robust, leading to a solid conclusion of using PIMs as adsorbents to remove dyes from water.

I have the following questions in mind that remain unanswered in the manuscript:

1) As surfactants are used here as well, are they being adsorbed into the PIMs as well?

2) Did the authors determine the pore size distribution of the PIMs used here before and after MD adsorption and desorption? This information will be crucial to show the location of these dyes as well as the effect of surfactants on adsorption. I understand that adsorption of dyes is nearly impossible without surfactants here, but will the surfactants be adsorbed into the PIMs as well? I think it will be good if the authors can provide some gas adsorption analyses on such samples to provide a conclusive finding for these questions.

3) Did the authors measure the dye concentration in the solution after ultrasonication of PIM-dye mixtures? Will the amount of dye released in this solution be equivalent to the amount dye removed from the original MB/water mixture?

Overall, I think this paper is well-written and should be published after addressing these concerns.

Reviewer: 2

Comments to the Author(s)

In this work, the author reported two spirobifluorene PIMs to remove the methylene blue (MB) dye from wastewater with the help of surfactants. The author explored the effects of different types and concentration of surfactants on the adsorption efficiency, and proposed the adsorption mechanism through reliable tests. The manuscript is well organized and has a great effect on the wastewater treatment. Therefore, I recommend this manuscript may be accepted for publication subjected to the incorporation for following minor corrections.

(1) Why does different chains on the benzene ring lead to the difference in adsorption performance, can the author simply explain this question from the structural perspective?

Author's Response to Decision Letter for (RSOS-200741.R0)

See Appendix A.

Decision letter (RSOS-200741.R1)

Dear Dr AlHetlani:

Title: Spirobifluorene-based polymers of intrinsic microporosity for the adsorption of methylene blue from wastewater: effect of surfactants

Manuscript ID: RSOS-200741.R1

It is a pleasure to accept your manuscript in its current form for publication in Royal Society Open Science. The chemistry content of Royal Society Open Science is published in collaboration with the Royal Society of Chemistry.

On behalf of the Subject Editor Professor Anthony Stace and the Associate Editor Dr Chaohua Cui.

RSC Associate Editor
Comments to the Author:
(There are no comments.)

Reviewer(s)' Comments to Author:

Appendix A

Dear Editor,

Thank you for your recent email regarding the decision on our submission to Royal Society Open Science RSOS-200741. We are very pleased with the decision and would like to take this opportunity to address the reviewer comments.

We hope that you find our responses satisfactory and that the revised version is now suitable for publication.

Reviewer comments to Author:

Reviewer: 1

Comments to the Author(s)

This is an interesting piece of work using state-of-the-art PIMs to remove dyes from water via physisorption. The characterisation techniques deployed here are complementary of each other. The language used here is suitable for scientific publication and the experiments here are very robust, leading to a solid conclusion of using PIMs as adsorbents to remove dyes from water.

I have the following questions in mind that remain unanswered in the manuscript:

1) As surfactants are used here as well, are they being adsorbed into the PIMs as well?

The surfactants used in this work all had a hydrophobic tail, which we believe is the part of the surfactant in contact with the polymers due to their hydrophobic nature. The other part of the surfactant (charged head) is in contact with the dye and this is responsible for the electrostatic interaction. So adsorption of the surfactants on the polymers took place to facilitate the dye adsorption.

2) Did the authors determine the pore size distribution of the PIMs used here before and after MD adsorption and desorption? This information will be crucial to show the location of these dyes as well as the effect of surfactants on adsorption. I understand that adsorption of dyes is nearly impossible without surfactants here, but will the surfactants be adsorbed into the PIMs as well? I think it will be good if the authors can provide some gas adsorption analyses on such samples to provide a conclusive finding for these questions.

Another interesting consideration from the reviewer. The pore size distribution on PIMs is typically history dependent. To measure it accurately after the adsorption of the surfactants, and due to the immersion of the PIM into a liquid, the polymer should be washed several times with various solvents and kept under vacuum, otherwise the measurement of the PSD would reveal complete loss of porosity independently from the adsorption or not of the surfactant into the pores. This washing typically resets the history of the sample, going back to the original pore size. Because of that, we assume that the proposed experiment would not tell us more on the potential effect, that the adsorption of the surfactant into the PIM has in terms of dye removal, and so it would not answer the reviewer question in a definitive manner.

3) Did the authors measure the dye concentration in the solution after ultrasonication of PIM-dye mixtures? Will the amount of dye released in this solution be equivalent to the amount dye removed from the original MB/water mixture?

Thank you for this good question. Ultrasonication of PIM-dye mixtures were performed to ensure that all the dyes are released from the polymer back into the solution so that the polymer can be filtered and collected for the reusability study. However, some of the polymer is lost during washing/filtration resulting in small loss in the adsorption efficiency in our case it was less than 8%. It would be interesting to perform such protocol in our future work using similar materials to account for the amount of dye removed from water.

Overall, I think this paper is well-written and should be published after addressing these concerns.

Reviewer: 2

Comments to the Author(s)

In this work, the author reported two spirobifluorene PIMs to remove the methylene blue (MB) dye from wastewater with the help of surfactants. The author explored the effects of different types and concentration of surfactants on the adsorption efficiency, and proposed the adsorption mechanism through reliable tests. The manuscript is well organized and has a great effect on the wastewater treatment. Therefore, I recommend this manuscript may be accepted for publication subjected to the incorporation for following minor corrections.

(1) Why does different chains on the benzene ring lead to the difference in adsorption

performance, can the author simply explain this question from the structural perspective?

The size and bulkiness of the side chains lead to different arrangements of the polymer chains and so to different pore size, depending on the bulkiness effect that they create. This different structural arrangement typically leads to size selectivity when used for gas separation. In the case of this paper, we demonstrated that it also leads to different performance for dyes removal when the polymer is dispersed into a solution.

Sincerely,

Entesar Al-Hetlani